# Factors associated with poor outcomes among people living with HIV started on anti-retroviral therapy before and after implementation of "test and treat" program in Coastal Kenya

Isaac Chome Mwamuye[1]*, Simon Karanja[1], Joseph Baya Msanzu[2], Aggrey Adem[2], Mary Kerich[1], Moses Ngari[3,4]

1 Jomo Kenyatta University of Agriculture and Technology, Mombasa, Kenya, 2 Technical University of Mombasa, Mombasa, Kenya, 3 KEMRI Wellcome Trust Research Programme, Kilifi, Kenya, 4 Department of Public Health, Pwani University, Kilifi, Kenya

* isachome@gmail.com

## Abstract

### Objectives

To determine the factors associated with poor outcomes among people living with HIV (PLHIV) started on anti- retroviral therapy before and after implementation of "Test and treat" program in 18 facilities in Coastal Kenya.

### Methods

A retrospective cohort study design was used to study PLHIV aged > 15 years and started on ART in the periods of April to August 2016, and April to August 2017, then followed up for 24 months. Primary outcome was retention defined as being alive and on ARVs after 24 months. Death and loss to follow-up were considered as poor outcomes. Kaplan–Meier survival methods were used to describe time to primary outcome. Cox proportional regression analysis was used to determine factors associated with poor outcomes.

### Results

86 patients (470 before test and treat, and 316 after test and treat cohorts) were enrolled. Overall, the median [IQR] age was 39.3 [32.5–47.5] years and 539 (69%) were female. After 24 months, retention rates for the before (68%) and after (64%) test and start groups were similar (absolute difference: -4.0%, 95%CI: -11-3.1, P = 0.27). There were 240(31%, 95%CI 27 to 34%) PLHIV with poor outcomes, 102 (32%) and 138 (29%) occurred among the test and treat group, and delayed treatment patients respectively. In multivariable regression model, test and treat had no significant effect on risk of poor outcomes (aHR = 1.17, 95%CI 0.89–1.54). Increasing age (aHR = 0.98, 95%CI 0.97–0.99), formal employ- ment (aHR = 0.42, 95%CI 0.23–0.76) and not being employed (aHR = 0.53, 95%CI 0.34–

**Data Availability Statement:** All relevant data are within the manuscript and its Supporting information files.

**Funding:** The author(s) received no specific funding for this work.

**Competing interests:** The authors have declared that no competing interests exist.

0.81) were negatively associated with poor outcomes. The risk of poor outcomes was higher among males compared to female patients (aHR = 1.37, 95%CI 1.03–1.82) and among divorced/separated patients compared to the married (aHR = 1.44, 95%CI 1.04–1.99).

## Conclusion

Retention patterns for the "test and treat" cohort were comparable to those who started ART before "test and treat". Patients who are males, young, divorced/separated, with poor socio-economic status had higher risks for poor clinical outcomes. Interventions targeting PLHIV who are young, male and economically disadvantaged provide an opportunity to improve the long-term outcomes.

## Introduction

Significant progress has been made in the fight against HIV and AIDS with over 25.4 million people living with HIV (PLHIV) on Anti-Retroviral Therapy (ART) out of the 38 million PLHIV globally [1]. Majority of the PLHIV are in Eastern and Southern Africa, accounting for 53% (19.6 million) of the global burden, with about 1.5 million being Kenyans [2]. By 2017, about 12.9 million (66%) PLHIV in the region were accessing antiretroviral therapy, among them 1.16 million Kenyans [3]. Also, by 2017, the estimated percentage of people living with HIV who achieved viral suppression in the region was 52%, and 56–60% in Kenya [1].

The uptake of ART accelerated in the recent 5 years due to increased access to ART as a result of the World Health Organization (WHO) guidance released in 2015 on when to start ART and Pre-exposure prophylaxis (PrEP) [4], for countries to treat all HIV infected people with Highly Active Anti-Retroviral drugs (HAART) irrespective of their $CD_4+$ levels or WHO stage. Kenya adopted the guidance in July 2016 with a campaign conducted to initiate ART to all the PLHIV who were in care but not started on ART [5]. This was implemented in the Coast counties from September 2016 after capacity building of health care workers and distribution of commodities. By March of 2017, all the clients in Mombasa, Kilifi and Kwale Counties who were on care had been started on ART, while newly identified PLHIV were immediately started on ART as per the new guidelines, as soon as they were identified and adequately prepared to continue with treatment. The new guidance was famously referred to as the "Test and treat" locally and required PLHIV to start ART immediately or within 14 days of HIV diagnosis. Prior to the "Test and treat", the National AIDS and STI Control (NASCOP) guidelines of 2014 [6] were in use which required only PLHIV who met one or more of the following criteria to be started on ART: pregnant or breastfeeding women, children below 10 years, people with $CD_4$ <500 cells/ml, WHO stage 3 or 4, TB/HIV con-infection and Hepatitis B co-infection.

Based on scientific evidence that was available then, a mathematical model done in 2009 on universal voluntary HIV testing with immediate antiretroviral therapy as a strategy for elimination of HIV transmission, showed that immediate ART for those identified as HIV positive would reduce HIV associated mortality to less than 1 case per 1000, and reduce HIV prevalence to less than 1% from the high of almost 5% in most Sub-Saharan Africa countries [7]. A prospective cohort study was done at 14 sites in 7 African countries (Botswana, Kenya, Tanzania, Uganda, Rwanda, South Africa, and Zambia) involving 3,381 sero-discordant couples who were followed up for a period of 24 months. In the cohort, 349 of the study participants who were HIV positive and eligible for ART were started on ART appropriately as per guidelines.

At the end of the study, out of the 103 new HIV infections among the previously sero-negative partners, only one was linked to a client who was on ART, demonstrating a 92% reduction in transmission rates in the group who were on ART [8]. In Uganda, a study among sero-discordant couples showed similar findings with marked reduction in HIV transmission with the use of ART from an incidence of 9.2/100 person-years to zero infections [9]. The TEMPRANO ANRS 12136 study and the Strategic Timing of Antiretroviral Treatment (START) study provided further important evidence to support universal ART by demonstrating better clinical outcomes in HIV asymptomatic patients who start ART at an early stage of their disease, when $CD_4$+ cell counts are above 500 cells per cubic millimeter [10, 11].

Although studies have demonstrated overwhelming evidence of the benefit of "Test and treat" on reducing HIV transmission [7], evidence on other outcomes like attrition from care have been inconsistence. The retention rate reported in a pooled analysis of 154 cohorts in Low- and Middle-Income countries was approximately 70%, two years after starting ART (2008 to 2013) [12]. Recent data after the universal test and treat policy suggest the retention has not improved, in Democratic Republic of Congo, the retention rate was 77% after two years of starting ART following the adoption of test and treat policy [13]. The same study found higher rate of attrition after the adoption of the universal test and treat policy. Another cohort in Masaka, Uganda, found PLHIV starting ARTs within seven days of HIV diagnosis had higher risk of lost to follow-up [14]. Systematic reviews report varying interventions to improve retention on care but with mixed outcomes [15]. Various studies have also reported different predictors of attrition for different follow-up times and in different settings, some before and others after the universal test and treat policy [13, 14, 16].

With the adoption of the "Test and treat", new evidence is thus needed on the retention patterns and identification of individual and health system factors that are associated with clinical outcomes. In this study, we evaluated the effectiveness of the "Test and treat" program in eighteen [17] facilities spread across three coastal counties (Mombasa, Kilifi and Kwale) of Kenya, and provided evidence on factors associated with poor clinical outcomes among PLHIV on ART.

## Materials and methods

### Study setting

This study was conducted in 18 HIV treatment centers in 3 coastal counties of Kenya; namely Mombasa, Kilifi, and Kwale. It involved the 6 centers with the highest number of clients started on ART in the April to August 2017 period in each of the counties, which are also referral centres for the counties and therefore gave a representation of PLHIV from all corners of the three counties. The three counties have a combined population of 3 million and, by time of study, the HIV prevalence was 4.1% in Mombasa and 3.8% for both Kilifi and Kwale(17), with majority of the population living below the poverty line.

### Study design

This was a retrospective observational cohort study in which attrition and retention for two cohorts of PLHIV was collected retrospectively, analyzed and compared. The first cohort included PLHIV who started ART in the period of April to August 2016 before the counties of Kilifi, Kwale and Mombasa implemented the WHO recommendations of universally treating all HIV positive people with Highly Active Anti-Retroviral Therapy (HAART). The second cohort included PLHIV started on ART in the period of April to August 2017 after the "Test and treat" guidance was implemented. The period between the two cohorts was excluded because it was assumed to be contaminated since the old clients who had not started ART

based on the recommendations of the old guidelines were being transitioned to the new guidelines.

## Study population

All PLHIV initiated on ART aged >15 years in the periods between April to August 2016, and April to August 2017 were recruited. Records documented in the ART registers formed the sampling frame for the study, which were 1623 and 1496 for before and after "Test and treat" respectively.

## Sample size determination

The sample size was estimated on the basis of statistical power to show significantly higher hazard of Lost to Follow Up (LTFU) among the "test and treat" cohort, compared to the delayed treatment cohort. In a South African cohort, the adjusted hazard ratio of LTFU among the test and treat patients compared to the delayed treatment was 1.58 [17] while the proportion of LTFU among HIV patients was 33.6% in Kilifi, Kenya [18] and 34% in Nigeria [19].

Assuming a LTFU of ~30%, a two-tailed alpha of 0.05, with statistical power >80%, a sample size of at least 600 HIV patients (300 in each cohort) was enough to show a 58% higher risk of LTFU among the HIV patients starting ARTs under test and treat policy (aHR 1.58) with 207 expected LTFUs. However, the study collected and analyzed data from 786 patients (316 for "test and treat" and 470 for delayed treatment) [20].

## Sampling techniques

Probability proportionate to size sampling was used where each facility contributed numbers proportional to the numbers started on ART in the study period. Then, within each facility, simple random sampling was used to select the sample.

## Study variables

In this study, the outcome was retention defined as a state where patients were known to be alive and receiving ART at the end of the follow-up period [16], in this case at every 3 months for 24 months after starting ART for both cohorts. All PLHIV who died or were lost to follow up (LTFU) were considered to have poor outcome. Attrition was construed to have occurred in both cohorts if a client discontinued taking ART for any reason, including death, loss to follow-up, and stopping ARV medications [16, 21]. Socio-demographic factors (age, gender/sex, and marital status), socio-economic factors (education level, occupational status) and clinical features (nutritional status, WHO stage, presence of opportunistic infections, ART regimen) were the independent variables in this study.

## Data collection tools and sources

Quantitative data was extracted through desk review from ART registers and patient files in the selected facilities using a data abstraction tool translated in the online Open Data Kit (Kobocollect©), downloaded and stored in Microsoft excel database then backed up externally.

## Ethical considerations

In accordance with the principles governing research involving human participants, this study ensured that respondents' ethical rights were upheld through submission, review and approval of the study proposal by the Pwani University Ethical Review Committee (ERC/MSc/032/

2020). Only anonymized data were extracted from patients' files. All data collected as part of this study was handled with utmost confidentiality.

## Statistical analysis

Study data were extracted from patients' records using standard questionnaire designed on Open Data Kit (Kobocollect©) and exported to STATA Version 16.1 (College Station, Texas 77845 USA) for analysis. Continuous variables were assessed for outliers by plotting visual aids like histograms, scatter plots and q-q plots for assessing normality. Outliers and illogical variables were flagged and corrected by checking correct values in patient records. Body Mass Index (BMI) was calculated as weight (Kg) divided by square of height in metres and grouped following WHO classification: <18.5, 18.5 to 24.9, 25.0 to 29.9 and ≥30.0. Data was assumed not to be missing at random, an extra category 'missing' was added to each variable to ensure all patients were included in the regression models.

Continuous variables were reported as mean (±SD) or median (with IQR), depending on underlying distribution. Categorical variables were reported as counts with their respective percentages. Categorical variables were compared using chi-square test or Fishers' exact test between the two cohorts while continuous variables were compared using non-parametric Wilcoxon Rank Sum Test. The study main exposure was a binary variable classified as patients who were diagnosed with HIV and started on ARTs before the policy of "test and treat" which was introduced in 2016 and those diagnosed from 2017 onwards. This was a cohort design with 24 months of follow-up after starting ART, hence the "test and treat" patients were those diagnosed and started on ART in the course of 2017 and followed up for 24 months ending in 2019. Other exposures explored in the regression analysis were demographic, socio-economic and clinical features at time of starting.

The study outcome was retention up to 24 months after starting ART. The retention rates in the study at all the time points were calculated as follows:

$$\% \, RT_t = (C_o - T_t - D_t - LTFU_t)/(C_o - T_t)$$

where $C_o$ is all patients initiated on ART in the cohort;

$T_t$ is all patients transferred out of care by time $t$;

$D_t$ is all patients who died by time t; and

$LTFU_t$ is all patients lost to follow-up by time t.

$\%RT_t$ will therefore be the proportion of all patients-initiated ART in the cohort who did not transfer out of care, are still alive and in care at time $t$. The retention rates were reported as proportions and the differences in all the outcomes between "test and treat" and delayed treatment patients compared using two-sample test of proportions and the absolute differences reported.

All PLHIV who died or were lost-to-follow-up (LTFU) were considered to have poor outcome. Time to poor outcome $t$ was defined from date of starting ARTs to date of the events or completing 24 months of follow-up for those who were actively on ARTs and on follow-up after 24 months. Probability distributions of each event during 24 months of follow-up were calculated using the Kaplan–Meier survival approach and compared between the groups (delayed and "test and treat" groups) using log-rank test. PLHIV who transferred out of the study area were right censored at the time of leaving the cohort. To explore the effect of "test and treat" versus delayed treatment on poor outcome, Cox proportional hazard regression was performed with the main exposure ("test and treat" or delayed treatment) and adjusted for confounders collected. The proportional hazard assumption was tested using Schoenfeld residuals method. To account for HIV treatment care and other unobserved heterogeneity across

the three counties (Kwale, Mombasa and Kilifi), shared gamma frailty Cox regression models were performed. A base model was run with the main exposure adjusted for age and sex with the three counties as random effect component in the shared gamma frailty Cox regression models. The final multivariable models included all other confounders collected at time of starting ART. $CD_4$ counts were excluded in the regression models because a large proportion of patients (>50%) had no $CD_4$ results at the time of starting ARVs. The measure of effect reported was adjusted hazard ratios and their respective 95% confidence intervals. Final multivariable discriminatory power was assessed using Area Under the Receiver Operating Characteristics curve (AUC).

We performed a sensitivity analysis including only the test and treat cohort PLHIV who were started on ARTs with fourteen days of HIV diagnosis as recommended by the policy. All the PLHIV in the delayed cohort were included in this sensitivity analysis.

Finally, retention was assessed in the ART programme after 24 months of starting ARTs versus the collapsed poor outcomes (deaths or LTFU). Transfer out were right censored at the time of leaving the cohort. A binary outcome was created; either being active on ARTs at the end of 24 months follow-up or having one of the poor treatment outcomes. Using the same approach as with other outcomes above, shared gamma frailty Cox regression models were run and reported as adjusted hazard ratios.

## Results

### Descriptive statistics of study participants

The study enrolled 786 patients with 470 (60%) being from the cohort before "test and treat" and 316 (40%) in the "test and treat" cohort. A total of 341 (44%) patients were from Kilifi County, 332 (42%) were from Mombasa County, and 113(14%) from Kwale County. Overall, the median (IQR) age was 39.3(32.5–47.5) years, 539 (69%) of the patients were females, and majority (423 or 54%) were married. Only 136 (17%) were economically dependent, 236 (30%) were unemployed and 321 (41%) had secondary level education (Table 1).

For clinical characteristics, approximately half of the patients had normal BMI (18.5 to 24.9), while 119 (15%) were underweight (BMI<18.5). A total of 732 (93%) patients were initiated on TDF/3TC/EFV as their starting regimen. Approximately two-thirds (67%) of the patients were classified as WHO stage I while only 2 (0.3%) were in WHO stage IV. Only 73 (9.3%) had ≥500 $CD_4$ cells/mm^3 while 279 (36%) had <500 $CD_4$ cells/mm^3. Overall, the CD4 count was not significantly different between the two cohorts (P = 0.06) including across the different WHO stages (Fig 1). There were only 62 (7.9%) patients with opportunistic infections; with 36 (59%) having TB, 9 (14%) with herpes simplex virus, 4 (6.3%) with sexually transmitted infections, 6 (9.4%) with bacterial infections and 7 (11%) with oral candidiasis. Overall, the median (IQR) days to starting ARVs after HIV diagnosis was 6 (0 to 118), it was 0 (0 to 8) among patients on test & treat compared to 34 (12 to 567) days among the delayed cohort (P<0.001). Among the 316 patients on test & treat cohort, 233/316 (74%) started ART within seven days of HIV diagnosis. (Table 1).

### Retention patterns

In the first three months after starting ART, the retention rates were 88% for the "test and treat" cohort and 84% for those started on treatment before "test and treat", the absolute difference being 4.2%. However, this difference was not significant (95% CI -0.8–9.2%, P = 0.10). After two years of treatment with ARVs, the retention rates declined in both the "test and treat" and delayed treatment groups to 64% and 68% respectively, absolute difference being -4.0%. This difference was also not significant (95%CI -11-3.1%, P = 0.27) (Table 2). The

**Table 1. Descriptive characteristics.**

| Characteristic | Total patients (N = 786) | Test and treat group (N = 316) | Delayed group (N = 470) | P-value |
|---|---|---|---|---|
| **Demographic characteristics** | | | | |
| Sex | | | | |
| Female | 539 (69) | 212 (67) | 327 (70) | 0.46 |
| Male | 247 (31) | 104 (33) | 143 (30) | |
| Age in years: median (IQR) | 39.3(32.5–47.5) | 38.9(32.0–46.4) | 39.4 (32.5–47.5) | 0.57 |
| Recruiting County | | | | |
| Mombasa | 332 (42) | 85 (27) | 247 (53) | <0.001 |
| Kwale | 113 (14) | 49 (16) | 64 (14) | |
| Kilifi | 341 (44) | 182 (58) | 159 (34) | |
| Marital status | | | | |
| Married | 423 (54) | 165 (52) | 258 (55) | 0.08 |
| Single | 142 (18) | 48 (15) | 94 (20) | |
| Divorced/separated/widowed | 221 (28) | 103 (33) | 118 (25) | |
| **Socio-economic characteristics** | | | | |
| Education level | | | | |
| None | 50 (6.4) | 24 (7.6) | 26 (5.5) | <0.001 |
| Primary | 279 (36) | 136 (43) | 143 (30) | |
| Secondary | 321 (41) | 117 (37) | 204 (43) | |
| Tertiary | 136 (17 | 39 (12) | 97 (21) | |
| Employment status | | | | |
| Self employed | 201 (26) | 89 (28) | 112 (24) | 0.02 |
| Informal employment | 84 (11) | 34 (11) | 50 (11) | |
| Not employed | 236 (30) | 106 (34) | 130 (28) | |
| Economic status | | | | |
| Independent | 250 (32) | 99 (31) | 151 (32) | 0.53 |
| Semi-independent | 400 (51) | 168 (53) | 232 (49) | |
| Dependent | 136 (17) | 49 (16) | 87 (19) | |
| **Clinical characteristics** | | | | |
| Body Mass Index (BMI) | | | | |
| <18.5 | 119 (15) | 59 (19) | 60 (13) | 0.04 |
| 18.5 to 24.9 | 390 (50) | 147 (47) | 243 (52) | |
| 25 to 29.9 | 120 (15) | 45 (14) | 75 (16) | |
| $\geq$ 30 | 64 (8.1) | 20 (6.3) | 44 (9.4) | |
| Missing | 93 (12) | 45 (14) | 48 (10) | |
| WHO Infection stage | | | | |
| Stage I | 527 (67) | 208 (66) | 319 (68) | 0.08 |
| Stage II | 193 (25) | 86 (27) | 107 (23) | |
| Stage III | 64 (8.1) | 20 (6.3) | 44 (9.4) | |
| Stage IV | 2 (0.3) | 2 (0.6) | 0 (0) | |
| Starting ART regimen | | | | |
| TDF/3TC/EFV | 732 (93) | 301 (95) | 431 (92) | 0.05 |
| AZT/3TC/NVP | 21 (2.7) | 4 (1.3) | 17 (3.6) | |
| TDT/3TC/NVP | 11 (1.4) | 6 (1.9) | 5 (1.1) | |
| Others* | 22 (2.8) | 5 (1.6) | 17 (3.6) | |
| CD$_4$ level before ART initiation[#] (cells/mm^3) Median (IQR) | 355 (172–514) | 308 (172–440) | 369 (173–558) | 0.06 |
| Number of adherence sessions before ART initiation | | | | |

(*Continued*)

**Table 1.** (Continued)

| Characteristic | Total patients (N = 786) | Test and treat group (N = 316) | Delayed group (N = 470) | P-value |
|---|---|---|---|---|
| ≤1 | 265 (34) | 125 (40) | 140 (30) | 0.02 |
| 2 | 129 (16) | 47 (15) | 82 (17) | |
| ≥3 | 280 (36) | 109 (35) | 171 (36) | |
| Missing data | 112 (14) | 35 (11) | 77 (16) | |
| Had opportunistic infection | 62 (7.9) | 29 (9.2) | 33 (7.0) | 0.21 |
| Days to starting ARVs after HIV diagnosis, Median (IQR) | 6 (0–118) | 0 (0–8) | 34 (12–567) | <0.001 |

*ABC/3TC/LPV/r, ABC/3TC/EFV, AZT/3TC/EFV, D4T/3TC/NVP and TDF/3TC/NVP,

#CD4 test were not systematically conducted for the "test & treat" group at starting ARVs because they were not required to decide when to start ARVS, therefore only 283 patients had CD4 data, 73/316 (23%) among those in test & treat group,

All results are N (%) unless where specified,

IQR; Interquartile range,

The P-values are from chi-square test or Fishers' exact test (where any n<5) for categorical variables and Wilcoxon Rank Sum Test for continuous variables.

retention rates for the two cohorts were not significantly different on any of the follow-up months (Fig 2).

## Patterns of outcomes

During follow-up for 1,144 person-years, 240/786 (31%, 95%CI 27–34%) PLHIV had poor outcomes (died or LTFUs), a rate of 211 (95%CI 186–240) poor outcomes per 1,000 person-years. Out of the 240 poor outcomes, 214/240 (89%) and 26/240 (11%) were LTFUs and deaths respectively. Of these 240 poor outcomes, 102/316 (32%) and 138/470 (29%) occurred among

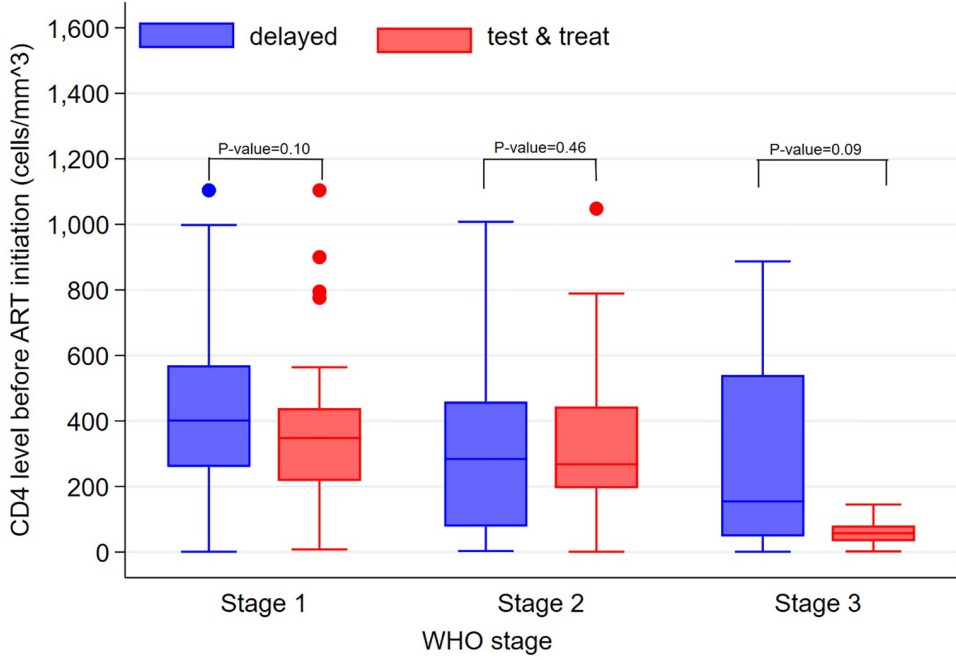

**Fig 1. Comparing CD4 count levels for test and start and delayed cohorts across the WHO stages.** *All the Patients in WHO stage IV were missing CD4 counts.

**Table 2. Cumulative retention rates from month 3 to 24 for "test and treat" and before "test and treat" cohorts.**

| Month | Test & start cohort (n = 316) | Before test and treat cohort (n = 470) | Absolute difference (95% CI) | P-value* |
|---|---|---|---|---|
| 3 | 269 (88) | 375 (84) | 4.2 (-0.8 to 9.2) | 0.10 |
| 6 | 243 (81) | 358 (81) | -0.2 (-5.9 to 5.6) | 0.95 |
| 9 | 237 (79) | 340 (77) | 1.6 (-4.5 to 7.6) | 0.62 |
| 12 | 228 (76) | 330 (75) | 1.1 (-5.2 to 7.4) | 0.74 |
| 15 | 210 (71) | 316 (72) | -1.4 (-8.1 to 5.2) | 0.67 |
| 18 | 204 (69) | 312 (71) | -2.3 (-9.1 to 4.5) | 0.50 |
| 21 | 191 (66) | 305 (70) | -3.9 (-10.9 to 3.0) | 0.26 |
| 24 | 183 (64) | 295 (68) | -4.0 (-11.1 to 3.1) | 0.27 |

*P-values from two-sample test of proportions

the 'test and treat' and delayed treatment patients: rates of 227 (95% CI 187–275) and 202 (95%CI 171–238) per 1,000 person-years respectively, age, sex and county adjusted HR 1.09 (95%CI 0.84–1.42) (Fig 3). Of the 240 poor outcomes, 52/240 (22%) occurred on the day of starting ART; with 23/102 (23%) in the 'test and treat' cohort and 29/138 (21%) from the delayed treatment cohort (P = 0.36).

**Individual level factors associated with poor outcomes.** In the multivariable regression model, "test and treat" had no significant effect on risk of poor outcomes (aHR = 1.17, 95%CI 0.89–1.54). However, increasing age was associated with protective effect on hazard of poor outcomes (aHR = 0.98, 95%CI 0.97–0.99). The hazard of poor outcomes was higher among male compared to female patients (aHR = 1.37, 95% CI 1.03–1.82). Compared to self-employed patients, those with lower risk of poor outcomes were the formally employed (aHR = 0.42, 95%CI 0.23–0.76) and not employed (aHR = 0.53, 95%CI 0.34–0.81). Divorced/separated patients had significantly higher hazard of poor outcome (aHR 1.44, 95%CI 1.04–

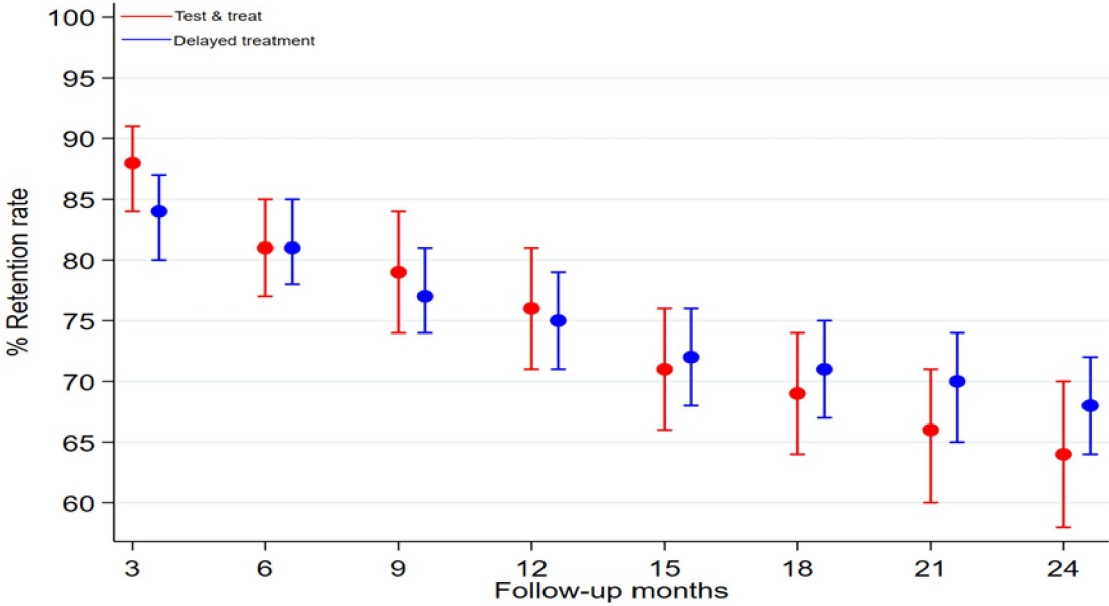

**Fig 2. Comparing retention rates at 3 months intervals for 24 months for cohorts before and after "test and treat" with their 95% confidence intervals.**

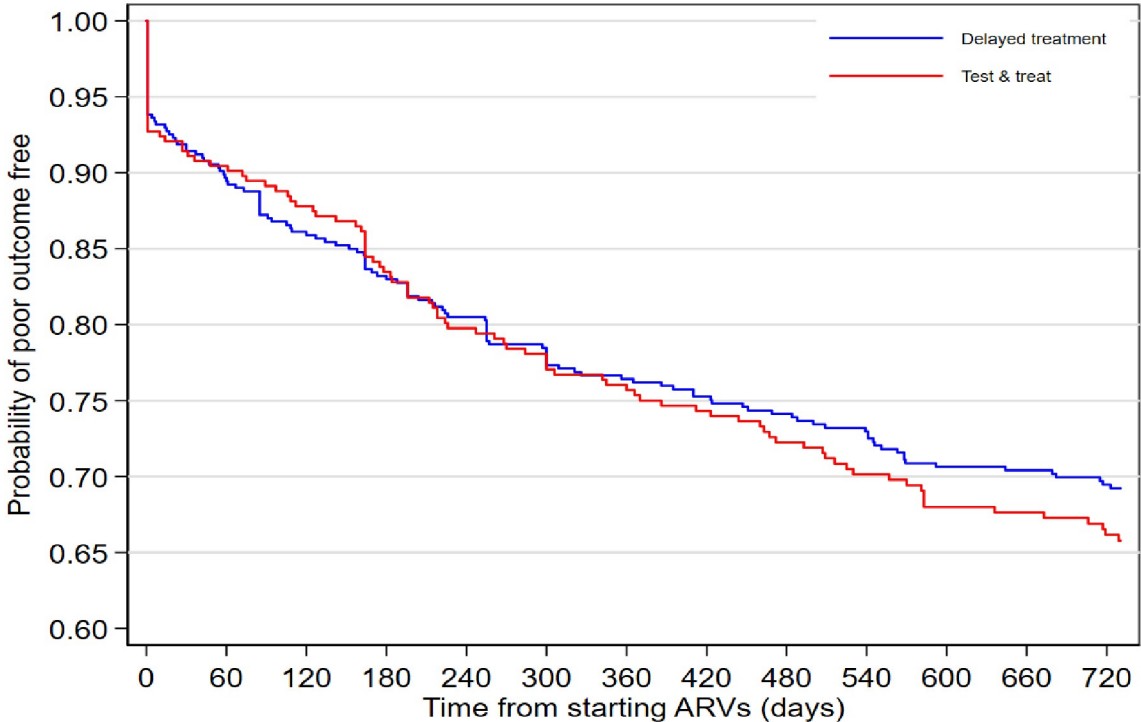

**Fig 3. Kaplan-Meier curve of not having poor outcomes for 24 months after starting ART.** The KM curve starts after one because of the poor outcomes that occurred at day zero.

1.99) compared to married patients. Other features explored were not associated with poor outcomes (Table 3). The multivariable regression model AUC was 0.66 (95%CI 0.62–0.70).

**Health system factors associated with poor outcomes.** Having regular facility staff meetings was associated with significant lower hazard of poor outcomes (aHR = 0.66, 95%CI 0.47–0.91, P = 0.01). The cadres of health-care workers involved in initiating clients on ART did not influence the clinical outcomes of PLHIV.

In the sensitivity analyses, including only the 233 PLHIV among the test & treat cohort who started ART within fourteen days of HIV diagnosis according to the guidelines, the retention rates at all the follow-ups were not significantly different (S1 Table). After adjusting for all exposures in Table 3, test & treat per policy was not associated with poor outcomes (aHR 1.22, 95%CI 0.90–1.66, P = 0.20). After adjusting for the health system variables on Table 4, test & treat per policy was not associated with poor outcomes (aHR 1.15, 95%CI 0.86–1.54, P = 0.35).

## Discussion

### Survivorship/Attrition patterns

In this study, the retention rates between the cohorts of PLHIV started on ART before and after implementation of test and treat were not significantly different. In both cohorts, the retention declined to a low of 64% and 68% among the test and treat and the delayed cohorts respectively after 24 months of starting ART. The retention at one year is comparable to that of 89% that was found in a 2017 study for test and treat patients in Kenya and Uganda [22]. A cohort study done in Uganda at almost the same time (January 2015 to December 2017) involving 646 patients concluded that "there was no significant difference (P = 0.231) in the mean retention times of patients initiated on ART based on $CD_4$ cell count compared to those

**Table 3. Multivariable analysis of individual level factors associated with poor outcomes.**

| Factors | Poor outcomes (N = 240) # | Adjusted HR (95% CI) * | P-value |
|---|---|---|---|
| *Type of treatment* | | | |
| Delayed treatment | 138 (29) | Reference | |
| Test and treat | 102 (32) | 1.17 (0.89–1.54) | 0.27 |
| *Age in years* | | | |
| <30 | 43 (18) | Reference | |
| 30 to 40 | 85 (35) | 0.84 (0.56–1.27) | 0.41 |
| 40 to 50 | 74 (31) | 0.70 (0.44–1.10) | 0.13 |
| ≥ 50 | 38 (16) | 0.54 (0.32–0.91) | **0.02** |
| *Sex* | | | |
| Female | 150 (28) | Reference | |
| Male | 90 (36) | 1.42 (1.07–1.88) | **0.02** |
| *Marital status* | | | |
| Married | 118 (28) | Reference | |
| Single | 46 (32) | 1.01 (0.69–1.48) | 0.97 |
| Divorced/separated | 67 (37) | 1.39 (1.01–1.92) | **0.04** |
| Widowed | 9 (23) | 0.98 (0.48–2.02) | 0.96 |
| *Education level* | | | |
| No school | 18 (35) | 1.51 (0.85–2.67) | 0.15 |
| Primary | 82 (30) | 0.91 (0.67–1.24) | 0.56 |
| Secondary & above | 140 (31) | Reference | |
| *Employment status* | | | |
| Self employed | 88 (33) | Reference | |
| Informal employment | 76 (38) | 1.23 (0.89–1.69) | 0.22 |
| Formal employment | 14 (17) | 0.41 (0.22–0.74) | **0.004** |
| Not employed | 62 (26) | 0.52 (0.34–0.80) | **0.003** |
| *Economic status* | | | |
| Independent | 74 (30) | Reference | |
| Semi-independent | 124 (31) | 1.10 (0.79–1.53) | 0.59 |
| Dependent | 42 (31) | 1.55 (0.93–2.58) | 0.10 |
| *BMI group* | | | |
| <18.5 | 46 (38) | 1.29 (0.90–1.85) | 0.16 |
| 18.5 to 24.9 | 115 (30) | Reference | |
| ≥ 25 | 44 (24) | 0.82 (0.58–1.18) | 0.29 |
| Missing | 35 (38) | 1.46 (0.95–2.23) | 0.08 |
| *Type of ART* | | | |
| TDF/3TC/EFV | 219 (30) | Reference | |
| Others** | 21 (39) | 1.39 (0.87–2.22) | 0.16 |
| *WHO stage* | | | |
| Stage I | 155 (29) | Reference | |
| Stage II | 59 (31) | 1.09 (0.78–1.54) | 0.60 |
| Stage III & IV | 26 (39) | 1.45 (0.86–2.45) | 0.16 |
| *Adherence to counseling sessions before ART initiation* | | | |
| ≤1 | 78 (30) | 1.06 (0.76–1.48) | 0.72 |
| 2 | 43 (33) | 1.24 (0.84–1.83) | 0.28 |
| ≥3 | 77 (28) | Reference | |
| Missing | 42 (38) | 1.79 (1.18–2.70) | **0.006** |

*(Continued)*

**Table 3.** (Continued)

| Factors | Poor outcomes (N = 240) # | Adjusted HR (95% CI) * | P-value |
|---|---|---|---|
| *Had opportunistic infection* | 19 (30) | 0.84 (0.47–1.50) | 0.56 |

*Adjusted HR from shared gamma frailty Cox model with the county as a random intercept

**ABC/3TC/LPV/r, ABC/3TC/EFV, AZT/3TC/EFV, D4T/3TC/NVP and TDF/3TC/NVP

# Results are N (%).

initiated under the "test and treat" strategy [23]. The retention rates are similar to other studies globally as was demonstrated by Fox & Rosen, who averaged retention to be 78% at 12 months, 71% at 24 months, and 69% at 36 months across all regions in a meta-analysis of 154 cohorts of PLHIV from 42 countries: 24 in Africa (114 cohorts), 10 in Asia (28 cohorts), and 8 in Latin American Countries (12 cohorts) [12, 24]. However, in a retrospective study in Malawi, immediate ART was found to be associated with low retention rates among PMTCT mothers. A multivariable analysis of 456 pregnant women on ART showed that "initiation of ART on the same day as HIV diagnosis, was independently associated with reduced retention in the first six months [25].

In this study, one in every ten cases of attrition were caused by death which is inconsistent with the study of Flynn et al., 2017 who observed death as the largest cause of attrition at 80% and 63% of the deaths which occurred in the first year of HIV therapy in a prospective cohort study in Uganda [26]. In our study, one in every five poor outcomes occurred on the day of starting ART and did not differ between the two cohorts, 23/102 (23%) for the test and treat

**Table 4. Multivariable analysis of health system factors associated with poor outcomes.**

| Factors | Poor outcomes N = 240 (%) | Adjusted HR (95%CI) * | P-value |
|---|---|---|---|
| *Type of treatment* | | | |
| Delayed treatment | 138 (58) | Reference | |
| Test and treat | 102 (43) | 1.14 (0.87–1.50) | 0.33 |
| *Cadre of health worker initiating ART* | | | |
| Medical officer | 4 (1.7) | 0.85 (0.30–2.40) | 0.76 |
| Clinical officer | 169 (70) | 0.81 (0.59–1.13) | 0.22 |
| Nurse | 114 (48) | 1.30 (0.90–1.86) | 0.16 |
| Counsellor | 86 (36) | 1.09 (0.73–1.62) | 0.67 |
| Peer educator | 40 (17) | 1.19 (0.78–1.82) | 0.41 |
| Community health volunteer | 23 (9.6) | 1.06 (0.66–1.71) | 0.80 |
| Mentor mother | 17 (7.1) | 0.98 (0.56–1.71) | 0.95 |
| Laboratory technician | 29 (12) | 1.41 (0.87–2.28) | 0.16 |
| Pharmaceutical technician | 68 (28) | 0.98 (0.63–1.52) | 0.93 |
| Health Records Information Officers | 74 (31) | 0.77 (0.53–1.13) | 0.19 |
| *Presence of regular staff meetings* | | | |
| Quality work improvement team meetings | 61 (26) | 0.71 (0.47–1.05) | 0.09 |
| Regular facility staff meetings | 119 (51) | 0.66 (0.47–0.91) | **0.01** |
| CCC departmental clinical meetings | 162 (70) | 0.83 (0.59–1.18) | 0.30 |
| Multi-disciplinary team meetings | 136 (59) | 1.03 (0.74–1.41) | 0.88 |

*Adjusted HR from shared gamma frailty Cox model with the county as a random intercept

# Results are N (%).

cohort and 29/138 (21%) for the delayed treatment cohort underlying the fact that starting patients on ART immediately upon diagnosis did not lead to poor outcomes. This underscores the need to do more to keep these patients on treatment. Thorough adherence counselling and treatment preparation should be prioritized when initiating ART to improve outcomes.

## Individual level factors associated with poor outcomes

Similar to previous studies, increasing age was significantly associated with protective effect on hazard of poor outcomes, with the risk of poor outcomes reducing by 2% for every year gained [27, 28]. This could be due to the reason that older people living with HIV are able to accept their HIV status and adhere to treatment, hence the better outcomes as opposed to adolescents who often struggle with self-stigma, relationships and self-identity issues.

Males had almost 40% higher likelihood of having poor outcomes compared to female patients. These findings are supported by other studies from Kenya, Uganda, Malawi and Nepal that found male gender to be associated with poor outcomes [29–32]. Among the reasons for the poor outcomes among men in Kenya are the poor health seeking behavior, less psychosocial support systems and higher levels of stigma compared to women. Divorced/separated patients were one and half times more likely to have poor outcomes compared to married patients which may be due to the lack of consistent support systems that may be lacking. This is in contrast to a study in Kilifi—Kenya [29] which did not find marital status to be associated with poor outcomes.

Compared to self-employed patients, formally employed patients had half the risk of having poor clinical outcomes, which is supported by other studies like the systematic review and meta-analysis led by Nachega which included 28 studies published between 1996 and 2014 involving 8743 HIV-infected individuals from 14 countries [33]. Inconsistent with our expectation, in this study, patients without any form of employment were found to have almost half the risk for poor outcomes when compared to those in self-employment. This is an area that requires further research to affirm and explain the findings.

This study found that patients who were economically dependent had more than 60% higher risk of poor outcomes compared to those who were economically independent. This is consistent with studies in Ethiopia [34] and Zambia [35] which concluded that "poor households are more likely to experience an AIDS death" and "low socioeconomic status in patients hospitalized for HIV/AIDS were more likely to die than high socioeconomic status inpatients" respectively. Such patients lack transport to access ART services, cannot afford nutritious food and other related services that would improve their clinical outcomes.

Patients who had missing documentation on the number of adherence counselling sessions before ART initiation had close to twice the risk for poor outcomes compared to those who had 3 or more counselling sessions. In most cases, patients without documentation of adherence did not receive any adherence counselling sessions at all. In a study among PMTCT clients in Ethiopia [36], women who received adherence counselling were more than 4 times likely to adhere to ART thereby leading to good clinical outcomes.

## Health system factors associated with poor outcomes

Among the health system factors analyzed, only having regular facility staff meetings was associated with significant lower hazard of poor outcomes. Regular staff meetings lead to coherence, stronger integration and team work among staff members. Team work through involving different staff cadres in a facility improved patient outcomes in a Chicago facility, USA [37]. However, the settings and staffing patterns are not similar to the Kenyan context, therefore, it may not be entirely applicable to the Coast of Kenya facilities. A Kenyan human

resources study in public service demonstrated that practices that enhance teamwork improved staff performance [38]. The type cadres of health care workers involved in imitating clients on ART did not influence the clinical outcomes of PLHIV.

## Strengths and limitations of the study

The strength of the study is the systematically collected data for 24 months after starting ART in a real-world setting. Although this study followed all the laid-out guidelines including the STROBE guidelines [39] for reporting, it still has a few limitations. First, the study relied on data in patient files and registers. CD4 counts were not systematically tested especially for the test & treat cohort who did not need this information to decide whether to start ART or not. It is therefore likely the CD4 results presented are subject to bias and thus the comparison made on Table 1 should be interpreted cautiously. Second, follow up for patients who were found to be lost to follow up was not done. In some studies, it is reported that some of the lost to follow up patients restarted treatment in other facilities and could be active on ART while others could have died. The study did not have the capacity to identify clients that had been diagnosed earlier, started ART and stopped, and later came back as newly diagnosed PLHIV. Lastly, since this is a retrospective study, bias arising from unmeasured features cannot be entirely ruled out.

## Conclusion and recommendations

Our findings suggest the "test and treat" program is as effective as the previous policy in retaining PLHIV on ARTs for at least 24 months after initiating ART. Interventions targeting PLHIV who are young, male and economically disadvantaged provide an opportunity to improve the long-term outcomes.

## Supporting information

**S1 Ethical clearance.**
(TIF)

**S1 Raw data.**
(CSV)

**S1 Table. Cumulative retention rates from month 3 to 24.**
(TIF)

## Acknowledgments

We would like to extend our gratitude to the research assistants in the 18 HIV treatment centers without whom this research work would not be complete. We also appreciate John Karisa, Meshack Mwangala and Ahmed Bunu who helped in coordination of data collection in the counties of Kilifi, Kwale and Mombasa counties.

## Author Contributions

**Conceptualization:** Isaac Chome Mwamuye.

**Data curation:** Isaac Chome Mwamuye, Moses Ngari.

**Formal analysis:** Isaac Chome Mwamuye, Moses Ngari.

**Investigation:** Isaac Chome Mwamuye.

**Methodology:** Isaac Chome Mwamuye, Simon Karanja, Joseph Baya Msanzu, Aggrey Adem.

**Project administration:** Isaac Chome Mwamuye.

**Supervision:** Isaac Chome Mwamuye, Simon Karanja, Joseph Baya Msanzu, Aggrey Adem, Mary Kerich.

**Validation:** Isaac Chome Mwamuye.

**Visualization:** Isaac Chome Mwamuye.

**Writing – original draft:** Isaac Chome Mwamuye.

**Writing – review & editing:** Isaac Chome Mwamuye, Simon Karanja, Joseph Baya Msanzu, Aggrey Adem, Moses Ngari.

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
