## [Decision Letter · Decision Letter 0]

17 Nov 2021

PONE-D-21-23151Factors associated with poor outcomes among people living with HIV started on anti-retroviral therapy before and after implementation of “Test and treat” program in Coastal KenyaPLOS ONE

Dear Dr. Mwamuye,

Thank you for submitting your manuscript to PLOS ONE. After careful consideration, we feel that it has merit but does not fully meet PLOS ONE’s publication criteria as it currently stands. Therefore, we invite you to submit a revised version of the manuscript that addresses the points raised during the review process.I recognize that your manuscript does not present new or innovative findings. Regardless, Plos One does not use this as a criteria for acceptance. Please address the reviewer's other comments to the best of your abilities.

We look forward to receiving your revised manuscript.

Kind regards,

Manish Sagar, MD

Academic Editor

PLOS ONE

2. Please provide additional details regarding participant consent. In the Methods section, please ensure that you have specified (1) whether consent was informed and (2) what type you obtained (for instance, written or verbal). If your study included minors, state whether you obtained consent from parents or guardians. If the need for consent was waived by the ethics committee, please include this information.

Reviewers' comments:

Reviewer's Responses to Questions

**Comments to the Author**

1. Is the manuscript technically sound, and do the data support the conclusions?

Reviewer #1: Partly

Reviewer #2: Partly

2. Has the statistical analysis been performed appropriately and rigorously? 

Reviewer #1: Yes

Reviewer #2: I Don't Know

3. Have the authors made all data underlying the findings in their manuscript fully available?

Reviewer #1: Yes

Reviewer #2: Yes

4. Is the manuscript presented in an intelligible fashion and written in standard English?

Reviewer #1: Yes

Reviewer #2: Yes

5. Review Comments to the Author

Reviewer #1: Summary and impression:

This was a retrospective cohort study PLWH who started on ART in the periods of April to August 2016 (delayed treatment group) compared to those in periods of April to August 2017 (test and treat group), each followed up for 24 months. Primary outcomes were death or loss to follow up.

The study enrolled from 3 coastal counties in Kenya and had a fair sample size, with the two comparison groups enrolled about a year apart; before and after local institutionalizing the WHO recommendation of test and treat in 2017. The participants for the cohorts were obtained from records in ART registers to identify those started on ART (470 enrolled) before and after (316 enrolled) Test and treat initiated. Primary endpoints were lost to follow up, viral suppression and death. Sample size well justified by earlier studies and adequate in this study, with total of 786 patients for analysis.

The main findings showed no difference in retention rate, viral load suppression rate and death in first 3 months and after 2 years of follow up. Although no difference in poor outcome between groups, individual factors were identified that were associated with increased risk of poor outcome in the test and treat group when compared to the delayed treatment. Higher risk with younger age, male gender, not employed and divorced status. No difference with the varied health care workers involved with care, only finding regular facility staff meetings associated did not seem to make sense. The overall conclusions were two-fold; retention patterns for the “test and treat” cohort was comparable to those who started ART before “test and treat”. Patients who are males, young, divorced/separated had higher risk of poor outcome in the test and treat group compared to the delayed treatment group.

Overall, the study addresses an important topic that can inform regional efforts on increasing ART adherence and HIV care retention in the test and treat era. However, the study is not novel and findings are not new; many previously studies have been done in other African countries.

The positives of this cohort study include the fact that this was real-life data as test and treat rolled out in the region. The comparison groups had fairly comparable distribution to further support the validity of these comparison groups. They were also well distributed among the counties in this region of Kenya, therefore, can make generalization in that entire region of Kenya and not just a particular township.

The significance of the finding is clear from this study and in the paper - that among HIV regardless of those with early treat or delayed, that poor outcome is associated with lack of support, less of economical independence, male gender and younger age. These are high risk groups which has and continues to need particular attention and focus in implementation research and public health efforts to increase adherence and retention in order to decrease risk of poor outcome and death. Tailoring efforts on these high risk groups of poor outcome is an important public health message.

However, the study is not novel nor innovative in its question and design. Several studies in African have looked at retention rate based on ART vs. test and treat already and stated in the paper so the presented data is not adding anything new to the existing body of literature on this topic, except that it confirms the high risk groups in this particular Kenyan region/country. There are some major and few minor improvements suggested in this current form.

1. Because the data is from an ART registry, it would be possible that those who enroll in the latter group (test and treat) could have been started on ART earlier, had some increase in CD4 and then stopped at some point before restarting again in this region where registry is located. These patients which I imagine from the overall low CD4 count of the groups would not be rare occurrences, but would be incorrectly categorizes as treatment naïve and also placed in the test and treat group at the later time period. But in fact, these patients should technically in the delayed group. The authors should attempt to determine how many participants had this scenario, because this would substantially bias the result and minimize the beneficial effects in the true test and treat. This would also explain why the test and treat group had lower CD4 count below 500 when one would expect that the CD4 T cell level be higher that the delayed treatment group as they are given the option of treating earlier before reaching CD4 <500.

2. Introduction focused on the merit and efficacy of test and treat in reducing transmission and severe disease. But this study is really focused on the topic of retention on ART and care, so authors should consider reviewing the literature on this specific topic; touching on the feasibility and barriers in implement and doing test and treat strategy.

3. Retention could have been affected by the ART regimen and the two groups differ in regimen. EFV may overall be better tolerated but could cause early discontinuation in some due to neurocognitive side effects and similar for NVP, with its own side effect profile. Would be of value to parse out if EFV vs NVP had any effect on retention.

4. I would expect that CD4 be lower in the delayed group but the CD4 is in fact lower in the test and treat group. This lower CD4 level, however, is consistent with the higher rate of OIs and the similar rate of poor outcome occurred on the day of starting ARVs in both groups (22% vs 23%). It seems like the study did not show difference in poor outcome because immunologically the patients were in similar states and similar CD4 count, albeit the test and treat was not significantly lower. It is unclear from the study design and warrant explanation in the discussion on why the test and treat had such low CD4 count.

5. It would be helpful to include P values in Table 1 with patient characteristics or mention in text to highlight any differences. From review of the data it does not seem like the two groups had any significant differences except maybe for the lower BMI categories (could be indicative of their state of health at enrollment) and maybe those who received NVP (could be factor in differences in retention rate).

Reviewer #2: The manuscript does not present any new data or information, as it has been shown from numerous studies that delaying HIV treatment leads to poor outcomes and countries worldwide have adapted the ‘Test and Treat’ strategy for treatment.

6. PLOS authors have the option to publish the peer review history of their article (what does this mean?). If published, this will include your full peer review and any attached files.

Reviewer #1: No

Reviewer #2: No

---

## [Author Response · Author response to Decision Letter 0]

25 May 2022

Reviewer 1.

Reviewer #1: Summary and impression:

This was a retrospective cohort study PLWH who started on ART in the periods of April to August 2016 (delayed treatment group) compared to those in periods of April to August 2017 (test and treat group), each followed up for 24 months. Primary outcomes were death or loss to follow up.

The study enrolled from 3 coastal counties in Kenya and had a fair sample size, with the two comparison groups enrolled about a year apart; before and after local institutionalizing the WHO recommendation of test and treat in 2017. The participants for the cohorts were obtained from records in ART registers to identify those started on ART (470 enrolled) before and after (316 enrolled) Test and treat initiated. Primary endpoints were lost to follow up, viral suppression and death. Sample size well justified by earlier studies and adequate in this study, with total of 786 patients for analysis.

The main findings showed no difference in retention rate, viral load suppression rate and death in first 3 months and after 2 years of follow up. Although no difference in poor outcome between groups, individual factors were identified that were associated with increased risk of poor outcome in the test and treat group when compared to the delayed treatment. Higher risk with younger age, male gender, not employed and divorced status. No difference with the varied health care workers involved with care, only finding regular facility staff meetings associated did not seem to make sense. 

Response:

Having regular staff meetings was taken to symbolize that the team had some level of teamwork and good leadership which could influence service delivery. 

The overall conclusions were two-fold; retention patterns for the “test and treat” cohort was comparable to those who started ART before “test and treat”. Patients who are males, young, divorced/separated had higher risk of poor outcome in the test and treat group compared to the delayed treatment group.

Overall, the study addresses an important topic that can inform regional efforts on increasing ART adherence and HIV care retention in the test and treat era. However, the study is not novel and findings are not new; many previously studies have been done in other African countries.

The positives of this cohort study include the fact that this was real-life data as test and treat rolled out in the region. The comparison groups had fairly comparable distribution to further support the validity of these comparison groups. They were also well distributed among the counties in this region of Kenya, therefore, can make generalization in that entire region of Kenya and not just a particular township.

The significance of the finding is clear from this study and in the paper - that among HIV regardless of those with early treat or delayed, that poor outcome is associated with lack of support, less of economical independence, male gender and younger age. These are high risk groups which has and continues to need particular attention and focus in implementation research and public health efforts to increase adherence and retention in order to decrease risk of poor outcome and death. Tailoring efforts on these high-risk groups of poor outcome is an important public health message.

However, the study is not novel nor innovative in its question and design. Several studies in African have looked at retention rate based on ART vs. test and treat already and stated in the paper so the presented data is not adding anything new to the existing body of literature on this topic, except that it confirms the high-risk groups in this particular Kenyan region/country. There are some major and few minor improvements suggested in this current form.

1. Because the data is from an ART registry, it would be possible that those who enroll in the latter group (test and treat) could have been started on ART earlier, had some increase in CD4 and then stopped at some point before restarting again in this region where registry is located. These patients which I imagine from the overall low CD4 count of the groups would not be rare occurrences, but would be incorrectly categorizes as treatment naïve and also placed in the test and treat group at the later time period. But in fact, these patients should technically in the delayed group. The authors should attempt to determine how many participants had this scenario, because this would substantially bias the result and minimize the beneficial effects in the true test and treat. This would also explain why the test and treat group had lower CD4 count below 500 when one would expect that the CD4 T cell level be higher that the delayed treatment group as they are given the option of treating earlier before reaching CD4 <500.

Response:

Thanks for highlighting this important observation. We extracted data from the patients’ files, we had no opportunity to interview the patients to get some of the data being requested. We agree with the reviewer this is a plausible scenario. However, the test and treat policy does not anticipate such scenario and to be pragmatic, the ART initiator relies on reported information from the patient. It may be challenging to establish a case restarting ARVs unless reported by the patient. Such patient would then not be treated as ART naïve because their date of HIV diagnosis would be recorded. We have acknowledged this as a study limitation in the second last paragraph of discussion section.

We have checked the dates of HIV diagnosis versus the date of starting ARTs to establish those who started ARTs within the seven days of diagnosis. We have added the days to starting ARTs on Table 1. Additionally, we conducted a sensitivity analysis including only the test & treat patients who started ARTs within seven days (74%). The results are shown on the last paragraph of results section and supplementary Table 1 below. 

Supplementary table 1. Cumulative retention rates from month 3 to 24 for “test and treat” and before “test and treat” cohorts.

Month Test & start cohort (n=233) Before test and treat cohort (n=470) Absolute difference (95% CI) P-value*

3 224 (87) 375 (84) 3.3 (-2.2 to 8.9) 0.25

6 172 (79) 358 (81) -2.3 (-0.9 to 4.3) 0.49

9 168 (77) 340 (77) -0.4 (-7.2 to 6.4) 0.91

12 162 (74) 330 (75) -0.9 (-7.3 to 5.5) 0.79

15 151 (69) 316 (72) -2.9 (-9.6 to 3.8) 0.40

18 145 (67) 312 (71) -4.1 (-10.9 to 2.7) 0.24

21 137 (66) 305 (70) -4.6 (-12.3 to 3.2) 0.24

24 131 (64) 295 (68) -4.9 (-12.8 to 3.0) 0.22

*P-values from two-sample test of proportions

In the test & treat policy, there is no systematic testing for CD4 counts because they are not needed to decide when to start ARTs as it used to be in the delayed cohort. Therefore, very few patients in the test & treat cohort have CD4 (~23%). These would be likely the very sick ones that a clinician feel the need to know their CD4 counts. It is highly likely the 23% CD4 counts we reported in this cohort is biased. We have checked if the CD4 counts were any different across the WHO stages stratified by the two cohorts and added these in the results section (figure 1). 

Figure 1: Comparing CD4 levels at starting ART for delayed and test and start cohorts

2. Introduction focused on the merit and efficacy of test and treat in reducing transmission and severe disease. But this study is really focused on the topic of retention on ART and care, so authors should consider reviewing the literature on this specific topic; touching on the feasibility and barriers in implement and doing test and treat strategy. 

Response:

Although studies have demonstrated overwhelming evidence of the benefit of “Test and treat” on reducing HIV transmission(7), evidence on other outcomes like attrition from care have been inconsistence. The retention rate reported in a pooled analysis of 154 cohorts in Low- and Middle-Income countries was approximately 70%, two years after starting ART (2008 to 2013)(12). Recent data after the universal test and treat policy suggest the retention has not improved, in Democratic Republic of Congo, the retention rate was 77% after two years of starting ART following the adoption of test and treat policy(13). The same study found higher rate of attrition after the adoption of the universal test and treat policy. Another cohort in Masaka, Uganda, found PLHIV starting ARTs within seven days of HIV diagnosis had higher risk of lost to follow-up(14). Systematic reviews report varying interventions to improve retention on care but with mixed outcomes(15). Various studies have also reported different predictors of attrition for different follow-up times and in different settings, some before and others after the universal test and treat policy(13,14,16).

3. Retention could have been affected by the ART regimen and the two groups differ in regimen. EFV may overall be better tolerated but could cause early discontinuation in some due to neurocognitive side effects and similar for NVP, with its own side effect profile. Would be of value to parse out if EFV vs NVP had any effect on retention. 

Response:

Among the PLHIV that we sampled, 95% (301/316) of the test and treat cohort and 92% (431/470) of the delayed treatment cohort were on EFV based regimen. We have formally explored the effects of the ART regimen on attrition in the multivariable regression model reported on Table 3. We found no evidence that attrition was affected by ART regimen. Even on Table 1 where we tested the differences using chi-square, the P-value indicates some borderline effect (P=0.05) which attenuated in the multivariable model. 

4. I would expect that CD4 be lower in the delayed group but the CD4 is in fact lower in the test and treat group. This lower CD4 level, however, is consistent with the higher rate of OIs and the similar rate of poor outcome occurred on the day of starting ARVs in both groups (22% vs 23%). It seems like the study did not show difference in poor outcome because immunologically the patients were in similar states and similar CD4 count, albeit the test and treat was not significantly lower. It is unclear from the study design and warrant explanation in the discussion on why the test and treat had such low CD4 count.

Response:

Following the test & treat policy, initiation of ARVs does not need a CD4 threshold. Therefore, most of these patients were not tested for CD4 counts (we have data for only 23%). It is likely the reported CD4 count among the test & treat could be systematically different from those not tested. However, we have compared the overall CD4 counts and across the WHO stages and found no evidence of differences between the test & treat versus the delayed cohorts (Table 1 and Figure 1). This is a general limitation of a study using data extracted from patient files. We have acknowledged this limitation on the second paragraph of the discussion section. 

We don’t have immunologically data to assess their differences between the two groups at baseline and during the follow-ups. Such data would be very useful to monitor the long-term effect of test & treat policy. We have mentioned this limitation and suggested future studies including ART program should systematically be collecting this data like the viral load.

5. It would be helpful to include P values in Table 1 with patient characteristics or mention in text to highlight any differences. From review of the data it does not seem like the two groups had any significant differences except maybe for the lower BMI categories (could be indicative of their state of health at enrollment) and maybe those who received NVP (could be factor in differences in retention rate).

Response: 

Thanks for pointing out this. We have updated Table 1 by adding the P-values, adding some footnotes to explain some variables especially the CD4 counts and added a new variable; time to starting ARVs from day of HIV diagnosis. The new variable highlights the difference in days to starting ARVs as would be expected. 

Reviewer 2

Reviewer #2: The manuscript does not present any new data or information, as it has been shown from numerous studies that delaying HIV treatment leads to poor outcomes and countries worldwide have adapted the ‘Test and Treat’ strategy for treatment.

Response:

We thank the reviewer for taking time to review our manuscript. We concur the test & treat policy has been rolled out worldwide. Our study is exploring the effectiveness of the test & treat post rolling out in a pragmatic setting. Although some data does suggest the policy is effectively, there are studies reporting the opposite. For example. our study did not find any differences in retention between the test & treat versus the delayed cohort. A more recent study in Masaka Uganda, reported higher lost to follow-ups among the test & treat compared to the delayed cohort (https://journals.plos.org/plosone/article?id=10.1371/journal.pone.0217606 ). There are also fears that asymptomatic patients with higher CD4 cell counts would have poor adherence to ART so that early initiation would results in loss to follow up (https://pubmed.ncbi.nlm.nih.gov/26148280/ ). It is also likely widespread use of antiretroviral treatment at a population and individual level may lead to development of drug resistance. 

There is thus need for more data from different settings assessing the long-term effect of the test & treat policy. Our study present data from different settings (to our best level of knowledge, no such study has been conducted across the three coastal counties of Kenya) and followed the PLHIV for a period of two years post ART initiation. Our findings that the retention was not any different between the two cohorts and that even the test & treat patients had attrition of more than one third after two years warrant more studies to understand the changing trends.

---

## [Decision Letter · Decision Letter 1]

15 Jun 2022

Factors associated with poor outcomes among people living with HIV started on anti-retroviral therapy before and after implementation of “Test and treat” program in Coastal Kenya

PONE-D-21-23151R1

Dear Dr. Mwamuye,

We’re pleased to inform you that your manuscript has been judged scientifically suitable for publication and will be formally accepted for publication once it meets all outstanding technical requirements.

Kind regards,

Manish Sagar, MD

Academic Editor

PLOS ONE

Additional Editor Comments (optional):

Reviewers' comments:

Reviewer's Responses to Questions

**Comments to the Author**

1. If the authors have adequately addressed your comments raised in a previous round of review and you feel that this manuscript is now acceptable for publication, you may indicate that here to bypass the “Comments to the Author” section, enter your conflict of interest statement in the “Confidential to Editor” section, and submit your "Accept" recommendation.

Reviewer #1: All comments have been addressed

Reviewer #2: (No Response)

2. Is the manuscript technically sound, and do the data support the conclusions?

Reviewer #1: Yes

Reviewer #2: Yes

3. Has the statistical analysis been performed appropriately and rigorously? 

Reviewer #1: (No Response)

Reviewer #2: Yes

4. Have the authors made all data underlying the findings in their manuscript fully available?

Reviewer #1: Yes

Reviewer #2: Yes

5. Is the manuscript presented in an intelligible fashion and written in standard English?

Reviewer #1: Yes

Reviewer #2: Yes

6. Review Comments to the Author

Reviewer #1: There were 5 recommendations for revisions and the authors have addressed them all. A few were beyond the scope of the data set reviewed but authors addressed these limitations in more detail. The study although not novel does add to the body of evidence showing the effectiveness of ‘test and Treat’ strategy and additional provides data on poor retention of this method, which seems to be a major concern. The effectiveness of this strategy does depend on the region of the world, the population and the socioeconimic status so this data shows data from this coastal area of Kenya.

Reviewer #2: The authors have addressed the comments from the reviewers and revised the manuscript accordingly.

The authors have amended the protocol and in the revised version they have:

- added in paragraph in introduction to discuss more about the background and HIV test and treat versus delayed treatment

- Updated Table 1 (with p-values)

- Updated the discussion on the study importance and value.

Hence, this revised draft of the manuscript appears to be scientifically suitable for publication.

7. PLOS authors have the option to publish the peer review history of their article (what does this mean?). If published, this will include your full peer review and any attached files.

Reviewer #1: No

Reviewer #2: No

---

## [Editor Report · Acceptance letter]

8 Sep 2022

PONE-D-21-23151R1 

Factors associated with poor outcomes among people living with HIV started on anti-retroviral therapy before and after implementation of “Test and treat” program in Coastal Kenya 

Dear Dr. Mwamuye:

I'm pleased to inform you that your manuscript has been deemed suitable for publication in PLOS ONE. Congratulations! Your manuscript is now with our production department. 

Kind regards, 

on behalf of

Dr. Manish Sagar 

Academic Editor

PLOS ONE